



# Departure from $K$-theory in the planetary boundary layer

Pedro Santos[1], Alfredo Peña[1], and Jakob Mann[1]

[1]DTU Wind Energy, Technical University of Denmark, Roskilde, Denmark

**Correspondence:** Pedro Santos (paas@dtu.dk)

**Abstract.** It is well known that when eddies are small, the eddy fluxes can be directly related to the mean vertical gradients, the so-called $K$-theory, but such relation becomes weaker the larger the coherent structures. Here, we show that this relation does not hold at heights relevant for wind energy applications. The relation implies that the angle ($\beta$) between the vector of vertical flux of horizontal momentum and the vector of the mean vertical gradient of horizontal velocity is zero, i.e., the vectors are

aligned. This is not what we observe from measurements performed both offshore and onshore. We quantify the misalignment of $\beta$ using measurements from a long-range Doppler profiling lidar and large-eddy simulations. We also use mesoscale model output from the New European Wind Atlas project to compare with the lidar-observed vertical profiles of wind speed, wind direction, momentum fluxes, and the angle between the horizontal velocity vector and the momentum flux vector up to 500 m both offshore and onshore, hence covering the rotor areas of modern wind turbines and beyond. The results show that within the

range 100–500 m, $\beta = -18°$ offshore and $\beta = -12°$ onshore, on average. However, the large-eddy simulations show $\beta \approx 0°$, partly confirming previous modeling results. We illustrate that mesoscale model output matches the observed mean wind speed and momentum fluxes well, but that this model output has significant deviations with the observations when looking at the turning of the wind.

## 1 Introduction

Current wind turbine rotors operate under vertical wind shear and wind veer conditions over a portion of the planetary boundary layer (PBL), which can go beyond the atmospheric surface layer (ASL) or "constant-flux" layer by tens or even hundreds of meters depending on the atmospheric stability and turbulence conditions. The Monin-Obukhov similarity theory (MOST) is used as the standard description of atmospheric turbulence within the ASL over flat and homogeneous conditions, e.g., far offshore (Businger et al., 1971). However, we still need a better understanding of turbulence beyond the ASL, which can aid at

improvements on the capabilities of numerical models to simulate the atmosphere. In particular, for wind energy, the Weather Research and Forecast (WRF) model is now widely used to simulate the dynamics of the atmosphere and the model outputs are continuously applied to produce wind climatologies, which are useful for wind resource assessment.

Recent studies have been taken advantage of measurements from long-range Doppler profiling wind lidars, which are able to probe the atmosphere up to ≈2 km, to gain insights on the wind climatology beyond the ASL, e.g., by characterizing the

vertical behavior of wind speed distribution parameters (Gryning et al., 2016), and by comparison of vertical wind profiles with those from mesoscale output using the WRF model (Floors et al., 2013). Regarding the estimation of second-order moments by





measurements from ground-based lidars, a variety of methods have been developed (Sathe and Mann, 2013). We are also now able to accurately compute the degree of attenuation and contamination of turbulence of lidars due to spatial averaging effects by the probe volume and scanning configuration (Mann et al., 2010; Sathe et al., 2011). Hence, lidar measurements beyond the

ASL provide us a unique opportunity to evaluate the ability of mesoscale models to reproduce turbulence characteristics and to improve the suite of parametrizations that such models offer. Some limitations in numerical models, i.a., WRF, come from hypothesis applied in the turbulence closures, which are responsible for well-known and long-standing biases when compared with observations (Brown et al., 2005; Sandu et al., 2013).

A number of PBL parameterizations in the WRF model are based on $K$-theory, in which the eddy fluxes are related to the

mean vertical gradients of velocity:

$$\langle u'w' \rangle = -K_m \frac{\partial U}{\partial z} \quad \text{and} \tag{1}$$

$$\langle v'w' \rangle = -K_m \frac{\partial V}{\partial z}, \tag{2}$$

where $U$ and $V$ are the two horizontal mean wind components, here aligned with the geographical coordinates, $u'$ and $v'$ are

fluctuations around both means, and $\langle \rangle$ represents the ensemble mean.

Each PBL scheme might formulate the eddy diffusivity or momentum exchange coefficient $K_m$ differently. As we move higher in the atmosphere, the influence of the Coriolis force grows, which results in a misalignment between the stress vector $(-\langle u'w' \rangle, -\langle v'w' \rangle)$ and the mean wind vector $(U, V)$. Further, a departure from $K$-theory might occur, which results in a misalignment between the stress vector and the vertical gradient of the mean wind vector $(\partial U / \partial z, \partial V / \partial z)$. Those angles are

hereafter referred to as $\alpha$ and $\beta$, respectively, illustrated in figure 1, and defined as:

$$\alpha = \text{atan2}(U, V) - \text{atan2}(-\langle u'w' \rangle, -\langle v'w' \rangle) \quad \text{and} \tag{3}$$

$$\beta = \text{atan2}\left(\frac{\partial U}{\partial z}, \frac{\partial V}{\partial z}\right) - \text{atan2}(-\langle u'w' \rangle, -\langle v'w' \rangle), \tag{4}$$

where $\text{atan2}(x, y)$ is defined such that it gives the angle of the vector $(x, y)$ relative to the $x$-axis.

Note that when numerical models use $K$-theory, $\beta = 0°$. Berg et al. (2013) showed non-zero vertical profiles of $\beta$, which were derived using observations from a short-range profiling lidar that measured up to 200 m at the onshore and rather flat site of Høvsøre, Denmark. Nonetheless, as mentioned by the authors, the surface heterogeneity might have impacted the results at Høvsøre.

In a study focused on low-level jets over open-water fetch, Svensson et al. (2019) showed a mismatch between lidar-observed

and WRF-derived along-wind momentum flux, where lidar second-order observations were filtered due to the effect of the





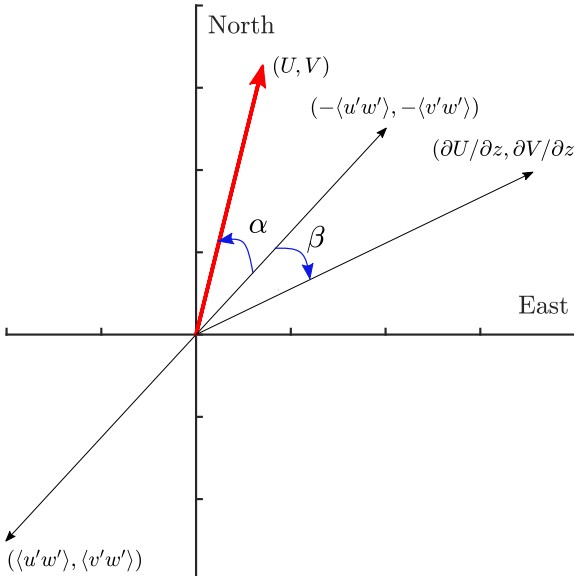

**Figure 1.** Definition of $\alpha$ and $\beta$ relative to the mean wind vector, the vertical gradient of the mean horizontal wind speed vector and the stress vector.

probe volume when compared to sonic anemometer measurements. However, they did not address the limitations of the numerical model with regards to the $K$-theory.

Using large-eddy simulation (LES) of the neutral, unstable, and stable PBL over flat and homogeneous terrain, Berg et al. (2013) found $\beta$ values close to zero throughout the PBL. Kosović and Curry (2000) performed an LES of the stable PBL and
found a maximum of $\beta \approx -10°$.

In this work, we compute mean wind and momentum flux profiles both at an offshore and an onshore location using both wind lidar measurements and the WRF model output from the New European Wind Atlas (NEWA) project (hereafter NEWA-WRF), in which a $K$-theory-based PBL scheme was the choice. The objectives herein are threefold:

1. The comparison of onshore and offshore vertical profiles of $\alpha$ and $\beta$ between lidar observations and the NEWA-WRF
output

2. Analysis of the behavior of $\beta$ as well as the wind direction bias between lidar observations and the NEWA-WRF output as a function of atmospheric stability

3. The evaluation of $\alpha$ and $\beta$ profiles using idealized LESs under offshore- and onshore-like conditions.

The preliminary study by Santos et al. (2020) found negative values of $\beta$ offshore. The present paper extends this work and
assesses the validity of $K$-theory, i.e., whether or not $\beta$ approaches zero, but using measurements both over land and offshore. Further, here we include the analysis of the vertical profile of $\beta$ using idealized LES both over land and offshore. We also discuss the consequences of the departure from $K$-theory for, e.g., mesoscale models.





## 2    Methodology

### 2.1    Site and measurements

We present results from two measurement campaigns, one offshore and another onshore, where a long-range pulsed profiling Doppler lidar (WindCube WLS70, Leosphere Inc., Saclay, France) was deployed next to meteorological masts for one-year measurement campaigns. Figure 2 shows the location of both sites: one at the FINO3 offshore research platform ($55°11.7'$N, $7°9.5'$E) and the other next to the Hamburg meteorological mast ($53°31.2'$N, $10°6.31'$E).

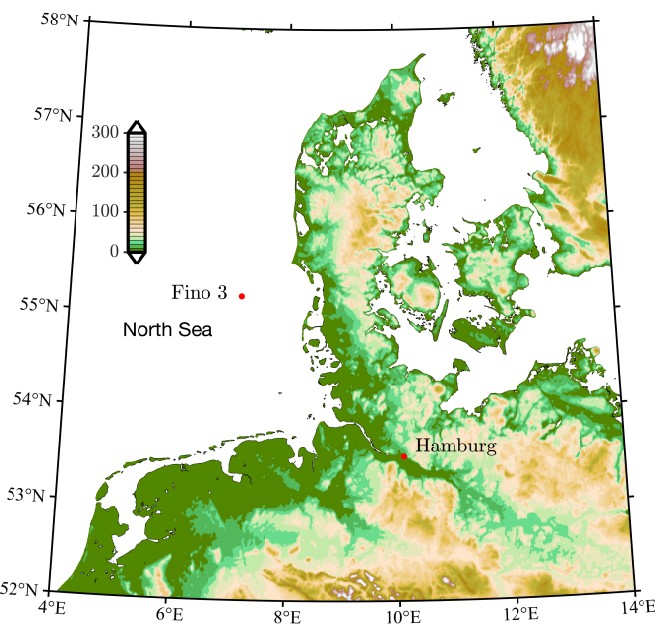

**Figure 2.** Map of northern Europe with the location of the offshore (FINO3) and onshore (Hamburg) sites as red dots. The colorbar indicate height above sea level.

The WLS70 measured the radial or line-of-sight velocity $v_r$ at four azimuthal positions, namely $v_{r,N}, v_{r,S}, v_{r,E}, v_{r,W}$, with

a half opening angle from the vertical of $\phi = 14.67°$. Pulses were sent to the atmosphere at each azimuth for more than 2 s and the wind vector can be thus derived every $\approx 10$ s within the range 100–2000 m above the lidar with a 50 m vertical resolution. The lidar probe volume length, given by the full width at half of the expected maximum (FWHM) of the lidar's along-beam sensitivity function, is $\approx 75$ m (Floors et al., 2013).

At FINO3, lidar measurements were performed from the platform's deck, which is 24.5 m above the mean sea level (amsl),

next to a fully instrumented 100-m meteorological mast. All heights are referred to amsl hereafter unless otherwise stated. A preliminary study detailed the instrumentation and showed the agreement between lidar measurements and a cup anemometer at 100 m for the first nine months of the campaign (Peña et al., 2015).


At the onshore site, the lidar was measuring for a one-year campaign 170-m away from a 280-m high television tower located southeast of downtown Hamburg. The tower is instrumented with METEK USA-1 sonic anemometers from Metek GmbH, Germany, at 50, 110, 175 and 250 m. Brümmer et al. (2012) showed a comprehensive introduction of the site and its wind climatology. Floors et al. (2013) used these onshore lidar measurements, among others, to assess the influence of baroclinicity on the measured wind profile.

## 2.2 Data selection and filtering

At FINO3, we select the entire lidar measurement period, between 1 September 2013 and 1 October 2014. We use concurrent 10-min measurements from a cup anemometer at 106 m to verify the lidar measurements. At Hamburg, we select the first two months of measurements from 6 April 2011 to 2 June 2011 where the lidar was operating without breaks and properly aligned with the north.

The lidar's carrier-to-noise ratio (CNR) is an indication of the quality of the $v_r$ estimation. For both sites, a CNR lower limit of $-29$ dB is used to filter radial wind speeds at all heights up to 500 m, so only fully available profiles up to this height are selected for the analysis. We choose $-29$ dB as a threshold since this value gives an unbiased estimation of the wind climatology (Gryning and Floors, 2019) and provides more valid profiles for robust statistics.

Along with the CNR threshold, a further filter was applied. This excludes data in which the radial velocity exceeds largely the median value, i.e., $|v_r - \tilde{v}_r| > 6\,\mathrm{m\,s^{-1}}$. This procedure allows to include periods with rain, which are usually compromised due to second returns from clouds and spikes in the time series.

Lidar measurements at 300 m had a consistently negative bias on the derived horizontal wind speed due to an interference in the laser signal, which was observed at both onshore and offshore campaigns. Hence, measurements at this height are removed in this analysis.

From the observations, we compute 30-min statistics throughout this work. From a sample of 10926 30-min periods at FINO3, we apply the above filters to get 76.2% of valid full profiles (8323) up to $500\,\mathrm{m}$. At the Hamburg site, 2719 30-min periods are filtered so we get 40% of valid full profiles (1089).

We compute the potential temperature gradient $\Delta\Theta = \Theta_{30m} - \Theta_{buoy}$, using concurrent air temperature measurements at 30 m and water temperature measurements at 6 m depth from the nearby meteorological buoy. Since there is no high frequency sonic anemometer data at FINO3, $\Delta\Theta$ is used as a simple proxy for atmospheric stability. Hence, $\Delta\Theta < 0$ K is used to identify unstable conditions and $\Delta\Theta > 0$ K for stable conditions.

At Hamburg, sonic anemometer measurements at 50 m represent surface-layer conditions (Floors et al., 2013). Therefore, we compute the Obukhov length ($L$) using these measurements and define stability regimes: $|L| > 500$ m represent neutral, $0$ m $\leq L \leq 500$ m stable, and $-500$ m $\leq L < 0$ m unstable conditions.

For FINO3, the wind lidar derived wind direction and a vane at $100\,\mathrm{m}$ on the meteorological mast showed a wind direction offset of $-11.7°$ (Peña et al., 2015). Thus, we apply this offset in our computations. At Hamburg, the lidar was aligned with the north for the selected period, so no offset was applied.





### 2.3 Lidar-derived momentum fluxes

The computation of momentum fluxes using a profiling lidar is based on the difference between the radial velocity variance $\sigma(v_r)$ of opposing lidar beams, i.e., of two opposing azimuthal measurements, originally proposed by Eberhard et al. (1989) and more recently applied by Mann et al. (2010),

$$
\begin{pmatrix} \langle u'w' \rangle \\ \langle v'w' \rangle \end{pmatrix} = \frac{1}{2\sin 2\phi} \begin{pmatrix} \cos\delta & -\sin\delta \\ \sin\delta & \cos\delta \end{pmatrix} \begin{pmatrix} \sigma^2(v_{r,E}) - \sigma^2(v_{r,W}) \\ \sigma^2(v_{r,N}) - \sigma^2(v_{r,S}) \end{pmatrix},
\tag{5}
$$

where $\delta$ is the angle between the north and the $v_{r,N}$ laser beam.

For this method, we assume statistical horizontal homogeneity as we combine radial wind speeds from a number of points in space. Therefore, we expect that the vertical profiles of momentum fluxes up to $500\,\mathrm{m}$ are better estimated using Eqn. (5) at the offshore than at the onshore site given the possible contamination from surface inhomogeneities at the latter.

### 2.4 Momentum fluxes from the NEWA-WRF outputs

NEWA-WRF outputs are available for all EU countries, including a $100\,\mathrm{km}$ offshore fetch plus all of the Baltic and the North Sea, and cover 30 yr (1989–2018). The dynamic forcing used was the $0.3°$ resolution ERA5 reanalysis. The detailed description of the model setup is given in Hahmann et al. (2020), which includes an evaluation of the performance of several PBL schemes available in the WRF model against measurements from several meteorological masts, i.a., FINO3. A further comparison of the NEWA-WRF outputs using more meteorological masts is given in Dörenkämper et al. (2020). Results of these comparisons demonstrated that a modified version of the Mellor-Yamada Nakanishi-Niino (MYNN) scheme (Nakanishi and Niino, 2009) resulted in the lowest wind speed bias at most sites.

The NEWA-WRF final product, i.e., a subset of the full model output, offers a 30-min time series over the entire 30-yr period, with a 3-km resolution and seven vertical levels at 50, 75, 100, 150, 200, 250, and 500 m. A linear interpolation was performed using the nearest neighbor grid cells to extract the time series at FINO3 and Hamburg from the model output over the spatial domain.

The MYNN level 2 (MYNN2) is a local scheme, i.e., $K_m$ is derived from local quantities. We can thus estimate the local momentum fluxes using Eqns. (1) and (2) from the NEWA-WRF output by computing first $K_m$ as

$$
K_m = lqS,
\tag{6}
$$

where $l$ is a master length scale, $q = \sqrt{2e}$ with $e$ being the turbulent kinetic energy (TKE), which in MYNN2 is based on the prognostic TKE from Mellor-Yamada (Mellor and Yamada, 1982), and $S$ is a non-dimensional eddy diffusivity coefficient, which accounts for atmospheric stability. Since $l$ is not stored in the NEWA-WRF output, we derive it following Nakanishi and Niino (2009).





## 2.5  Idealized WRF-LES

Previous LESs under neutral, stable, and neutral PBLs performed with a surface condition typical of an onshore site showed $\beta = 0°$ (Berg et al., 2013). We therefore performed two idealized LESs of a neutral PBL over a flat surface; one over water, i.e., a roughness length of 0.0002 m, and the other over land, i.e., a roughness length of 0.65 m, which is the value found over the urban sector at the Hamburg tower (Gryning et al., 2007). The LESs were performed using the WRF model (version 4.1.2) with a single domain of $7500 \text{ m} \times 7500 \text{ m} \times 2000 \text{ m}$ in the two horizontal and vertical directions, respectively, with a horizontal resolution of 15 m and a Coriolis parameter correspondent to the latitude of the FINO3 site and the Hamburg tower for the offshore and onshore simulations, respectively. The vertical grid spacing of $\approx$5 m was kept constant up to 250 m and stretched out thereafter. We used a time step of 0.2 s and the subgrid-scale model of Deardorff (1980).

The simulations were performed for a dry atmosphere. The initial potential temperature was kept constant (289.5 K) up to 700 m, where we imposed an inversion of 10 K km$^{-1}$. At all vertical levels, the initial $u$ and $v$ velocities were set to 11 and 0 m s$^{-1}$, respectively, at FINO3 and to 10 and 0 m s$^{-1}$, respectively, at Hamburg to try to match the highest observed values at both sites. MOST was applied at the surface via a WRF-in-built surface-layer scheme. The LES was run for 12 h, a heat flux of 0 K m$^{-1}$ was imposed at the surface, and periodic boundary conditions were applied.

The relevant parameters within the whole domain were output every 10 s within the period 10.5–11.5 h because we found the highest velocity for both simulations within this hour. The statistics were computed over this one hour period (such as means, variances, and covariances) and then spatially averaged over the whole domain.

The total momentum fluxes were computed by adding the subgrid to the resolved parts. Note that the coordinate system, in this case, is relative to the wind direction at the PBL top.

## 3  Results

The FINO3 and Hamburg sites are only separated by 267 km, hence they are influenced by the same large-scale weather patterns. At both sites westerly winds are prevalent (Brümmer et al., 2012; Peña et al., 2015); therefore, we select for the analysis the wind directions $270° \pm 45°$ and $275° \pm 45°$ for FINO3 and Hamburg, respectively, based on the measurements from the first vertical level of the lidar. At FINO3, this sector represents one with a several hundreds kilometers fetch, whereas at Hamburg, this is the 'urban' sector, which is characterized by industrial buildings and warehouses (Brümmer et al., 2012). The selected sectors represent $\approx 50\%$ of the amount of winds at FINO3 and $\approx 45\%$ for Hamburg. The size of the selected sectors does not have a significant impact on the vertical profiles presented below.

Figures 3 and 4 show the vertical profiles computed from the lidar measurements (blue lines) and concurrent NEWA-WRF results (red lines) within the westerly sector of FINO3 and Hamburg, respectively. For each vertical level, the shaded area represents the standard error of the mean given by $\pm \sigma / \sqrt{N}$, where $\sigma$ is the standard deviation and $N$ the number of observations. The mean NEWA-WRF's PBL height is 925 m at FINO3 and 858 m at Hamburg. Note that during stable conditions the profiles are close to the PBL top.





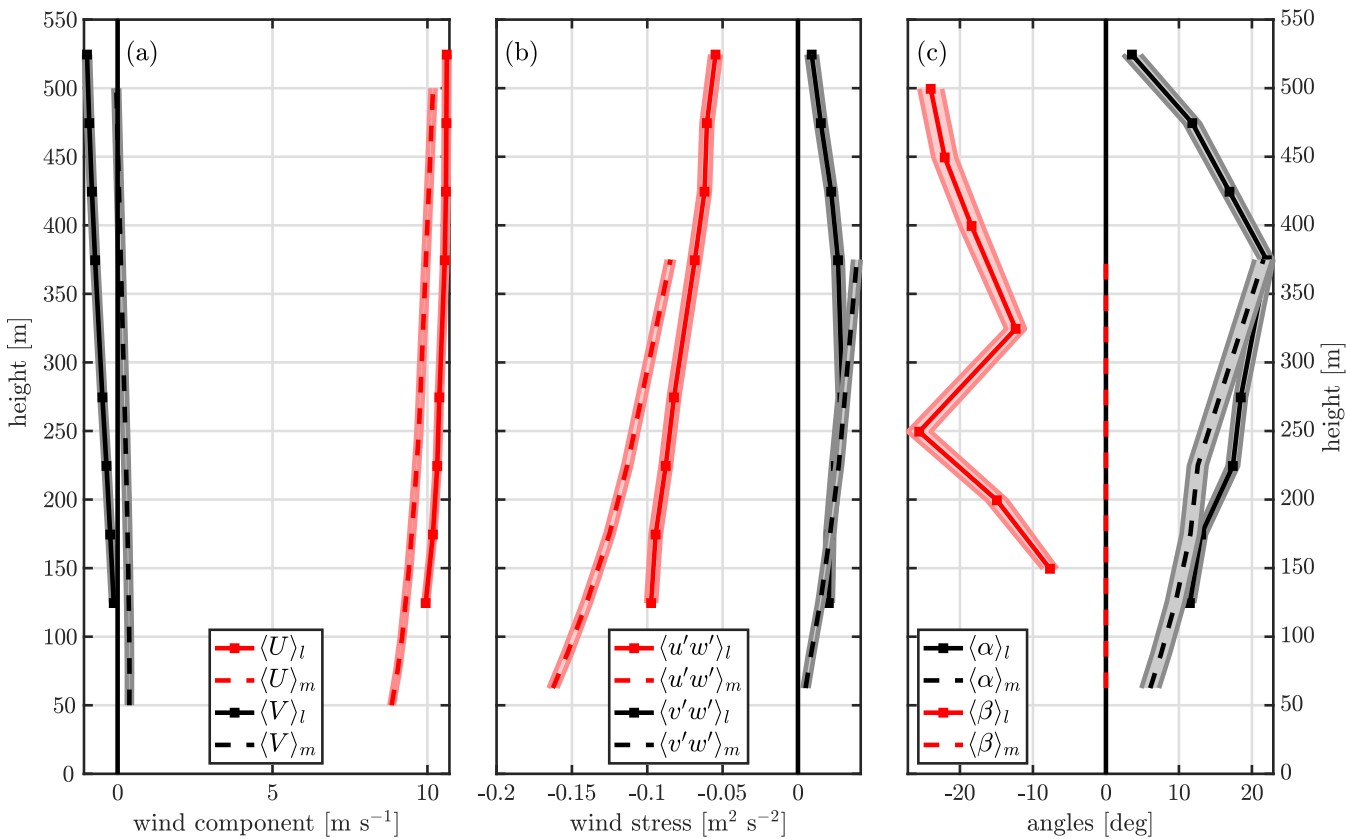

**Figure 3.** Mean offshore vertical profiles of (a) horizontal wind components, (b) momentum fluxes, and (c) turning angles for westerly winds ($270° \pm 45°$). Full lines represent lidar observations (subscript $l$) and dashed lines are NEWA-WRF (subscript $m$). The total number of 30-min profiles is 3343. Shaded areas denote associated standard errors of the mean.

### 3.1 Bias on wind turning

From the $U$ and $V$ vertical profiles at FINO3 (Fig. 3a), it is clear that NEWA-WRF underestimates the wind turning in the marine PBL. The fact that the modeled wind is veered (rotated clockwise) relative to the observed wind is a well-known issue from numerical weather models (Brown et al., 2005) and was already observed with long-range lidars in other sites (Peña et al., 2014). The wind turning is also not properly modeled at the Hamburg site (Fig. 4a), where NEWA-WRF shows that the modeled wind is backed (rotated anticlockwise) relative to the observations. Note that this type of pulsed Doppler lidar is ideal to quantify the turning of the wind, since it is capable to measure all vertical levels at once.

Figure 5 shows the vertical profiles of the wind direction bias between the lidar and NEWA-WRF for stable and unstable conditions at both sites. NEWA-WRF consistently underestimates the observed wind veer, with the highest bias under stable conditions, as expected (Brown et al., 2005). The difference between onshore (Fig 5a) and offshore (Fig 5b) conditions is that over water the wind direction bias is small ($\approx 1°$) at 100 m and increases with height. Contrarily, over the urban fetch the bias





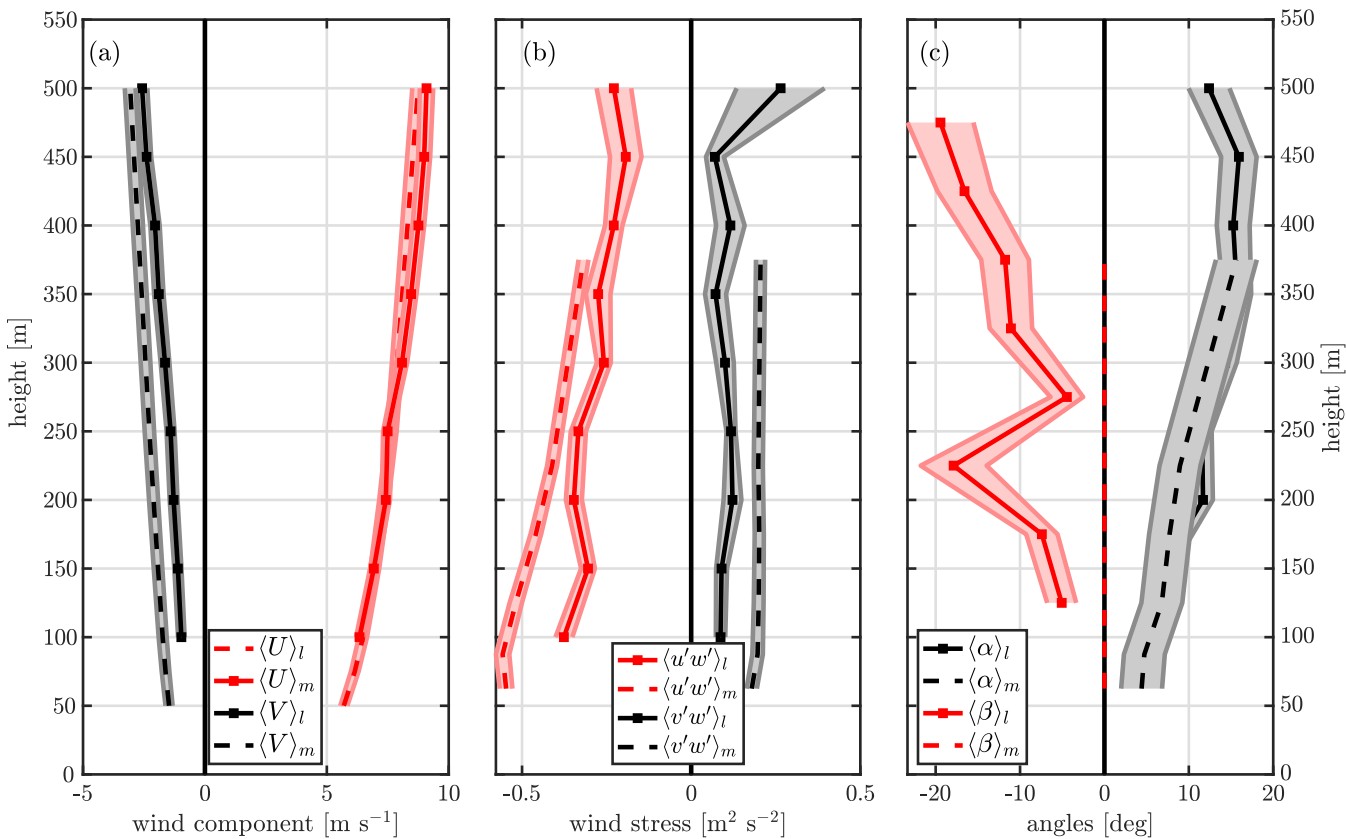

**Figure 4.** Mean onshore vertical profiles of (a) horizontal wind components, (b) momentum fluxes, and (c) turning angles for westerly winds (275°±45°). Full lines represent lidar observations (subscript $l$) and dashed lines are NEWA-WRF (subscript $m$). The total number of 30-min profiles is 616. Shaded areas denote associated standard errors of the mean.

is quite significant ($> 5°$) at 100 m and decreases towards the PBL top, where it reaches similar values to those at offshore conditions. The wind turning bias behavior under stable conditions is in line with previous findings for offshore conditions, where PBL schemes seem to produce more mixing which leads to an underestimation of both wind shear and veer (Simpson et al., 2018).

One possible cause for the wind direction bias observed in both sites would be if the model is systematically producing a distinct stability regime. NEWA-WRF also outputs $L$, so we can assess this assumption by comparing the concurrent modeled and observed stability. According to our atmospheric stability criterion, unstable conditions are predominant at FINO3 and represent 85.5% of the vertical profiles when considering all wind sectors. The mean NEWA-WRF PBL height for unstable conditions is 993 m. When comparing its sign with that from the measured potential temperature gradient, we find that 76% of the observed stable profiles are indeed simulated as such, whereas the number increases to 90% for the observed unstable profiles.





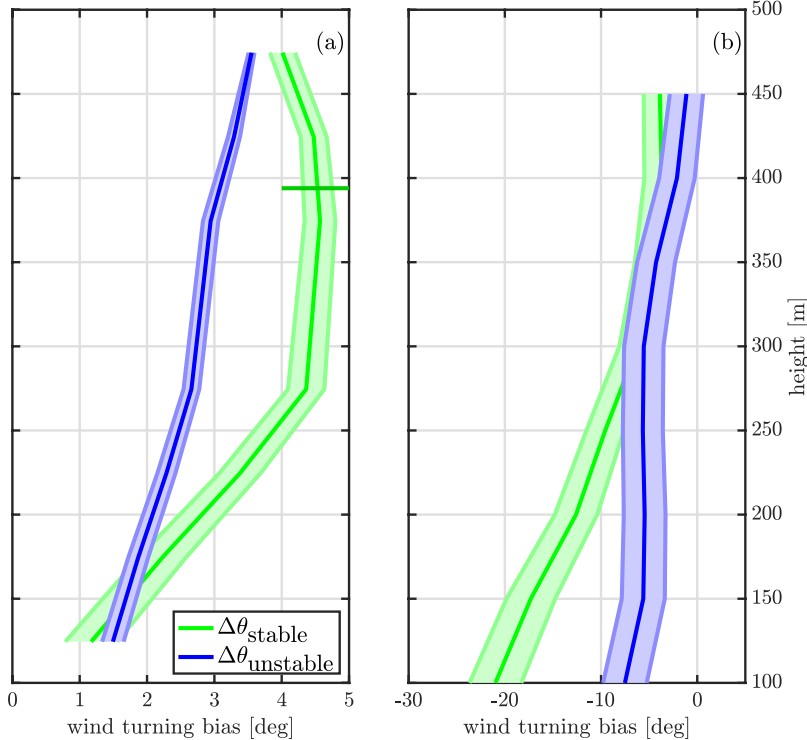

**Figure 5.** Vertical profiles of wind direction bias between the lidar observations and the NEWA-WRF output at FINO3 (a) and Hamburg (b) under unstable (blue) and stable (green) conditions. The horizontal green line represents the mean PBL height based on the NEWA-WRF output under stable conditions.

Under onshore conditions over the urban sector at Hamburg, neutral conditions are predominant, as discussed by Gryning et al. (2007). For all stability conditions at Hamburg, the estimated PBL height from NEWA-WRF is above the profiles (Fig.

5b). When $L$ computed by the sonic anemometer is compared to the output from NEWA-WRF, we find that 60% of the observed stable profiles are indeed simulated as such, whereas the number increases to 94% for the observed unstable profiles.

### 3.2  Momentum fluxes and lidar's turbulence attenuation

The vertical profiles of momentum flux (Figs. 3b and 4b) behave as expected for both sites given the analyzed westerly sectors, i.e., $\langle v'w' \rangle$ is close to zero and $\langle u'w' \rangle$ decreases in magnitude almost linearly with height both for NEWA-WRF and the

lidar observations. At both sites, there is however an important mismatch between the lidar observations and NEWA-WRF for $\langle u'w' \rangle$. This can be caused either by the method used to derive these fluxes based on the WRF model output or turbulence attenuation issues of the lidar observations (or both). The computed vertical profiles of velocity gradients are similar for both the NEWA-WRF output and the lidar observations (not shown). Therefore, if the bias comes from the NEWA-WRF estimations





mainly, then our estimate of $K_m$ is under suspicion. On the other hand, these lidar-derived momentum fluxes are inherently

biased due to spatial averaging along the lidar beam, i.e., probe-volume effects.

We perform an estimation of the lidar filtering effect under neutral conditions by considering that turbulence can be described by the model of Mann (1994) (hereafter Mann model) using a turbulent eddy-lifetime parameter $\Gamma = 3.9$, as recommended by the IEC (IEC, 2005). $\Gamma$ describes the degree of anisotropy of turbulence; $\Gamma = 0$ corresponds to fully isotropic turbulence. Figure 6 shows the ratio of the filtered to the unfiltered momentum flux $\langle u'w' \rangle_f / \langle u'w' \rangle_u$ as a function of the ratio of the lidar's probe

volume (i.e. the FWHM) to the turbulence length scale ($L_{MM}$). $L_{MM}$ is another parameter of the Mann model.

The black full line in Fig. 6 shows the prediction of $\langle u'w' \rangle_f / \langle u'w' \rangle_u$ using the Mann model. For a detailed description of the calculations, refer to Mann et al. (2010). For the WLS70 used in this study, FWHM = 75 m, so we only need to know $L_{MM}$ to compute the degree of filtering. We can estimate the turbulence length scale using the approximation by Kelly (2018),

$$L_{MM} \approx \frac{\sigma_{U_h}}{\partial U_h / \partial z},\tag{7}$$

where $\sigma_{U_h}$ is the standard deviation of the horizontal velocity and $\partial U_h / \partial z$ is the horizontal wind shear, which is computed here using the polynomial by Högström (1988) accounting for the observations at all vertical levels at each mast. The length scale at 100 m is computed as a linear interpolation between the closest measurements at each mast.

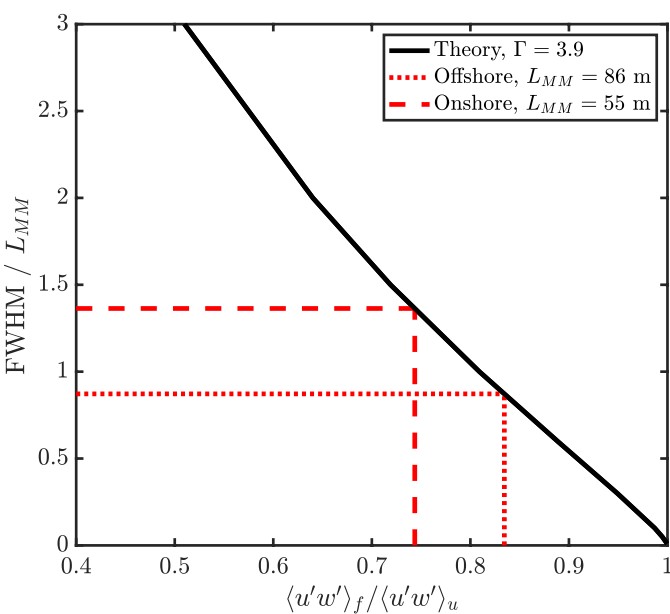

**Figure 6.** Ratio of the filtered to unfiltered momentum flux as a function of the ratio of the lidar's probe length to the turbulence length scale ($L_{MM}$). The dotted lines represent estimated values for onshore and offshore sites at 100 m.

The dashed lines in Fig. 6 show that the turbulence length scale estimated at FINO3 is larger than that at Hamburg, which

results in a larger attenuation of the momentum flux at Hamburg (74.4%) compared to that at FINO3 (83.4%). As turbulence





length scales depend on atmospheric stability (Peña, 2019), we expect that the dominant unstable conditions at FINO3 result in larger turbulence length scales compared to those at Hamburg where neutral conditions are more often observed.

By compensating the lidar-observed $\langle u'w' \rangle$ values at 100 m with the results in Fig. 6, we find a better agreement between the lidar- and NEWA-WRF derived values. However, its worth noticing that the NEWA-WRF momentum fluxes are parameterized
from Eq. (6) so it does not necessarily have the same magnitude as the observations. Furthermore, the attenuation estimated in Fig. 6 is subject to uncertainties from the spectral model, the FWHM value and the $L_{MM}$ estimation (refer to Appendix A for the vertical profiles of estimated $L_{MM}$ at each site).

### 3.3    The angles of the stress vector with mean wind vector ($\alpha$) and with mean wind shear vector ($\beta$)

Figures 3c and 4c show the vertical profiles of $\alpha$ and $\beta$ for offshore and onshore conditions, respectively. As expected from the
LESs of Berg et al. (2013), $\langle \alpha \rangle$ is close to zero near the surface and increases with height. Furthermore, the results using the NEWA-WRF output agree well with those of the lidar at both sites. However, note that a good agreement on $\langle \alpha \rangle$ can be due to a combination of biases in both wind direction and momentum fluxes.

In this study, the most evident mismatch between the lidar measurements and the NEWA-WRF output is with regards to $\langle \beta \rangle$, since we use $K$-theory to derive the eddy fluxes with the NEWA-WRF output. This theory assumes this misalignment is zero.
The misalignment $\langle \beta \rangle$ for all heights is on average close to $-18°$ offshore and $-12°$ onshore. If the wind sector decreases to $\pm 30°$, the mean $\langle \beta \rangle$ for all heights is $-17°$ offshore and $-9°$ onshore, i.e., the $\langle \beta \rangle < 0$ result is not sensitive to the wind sector in neither of the sites.

The $\beta$ profiles and, hence, the validity of $K$-theory are sensitive to the vertical gradients of the mean wind. Therefore, we expect that PBL schemes using the assumption of $K$-theory (such as MYNN2 and other local PBL schemes) show better
performance under stable conditions, i.e., where the vertical mean wind gradients are large (Shin and Hong, 2011). Figure 7 shows the vertical profiles of $\beta$ as a function of atmospheric stability for both sites. To avoid the spikes seen in Figs. 3 and 4, the third measurement height is shown here as a linear interpolation.

The vertical profiles of $\langle \beta \rangle$, derived from the lidar observations, show a lower value $\approx -7°$ on average for offshore and a close to $0°$ onshore under stable conditions. Additionally, under stable conditions $\langle \beta \rangle$ approaches zero close to the PBL top
under stable stratification. Although fluctuating, the $\langle \beta \rangle$ values are always larger under unstable than stable conditions at both sites.

### 3.4    WRF-LES

Figure 8 shows the WRF-LES results with regards to the turning angles for a neutral PBL over a water-like and a land surface roughness. Over water and below 250 m (Fig. 8a), $\alpha$ is negative due, partly, to the higher impact of the subgrid-scale model on
the $\langle u'w' \rangle$ term compared to the $\langle v'w' \rangle$ term. Within the range 250–550 m, $\alpha$ is comparable to the observed values at FINO3. $\beta$ is nearly $0°$ within a large portion of the PBL and increases (in magnitude) with height at the vertical level where both velocity gradients and eddy fluxes are near zero. Within the range $\approx 40$–200 m, $\beta$ becomes slightly positive because of the combination





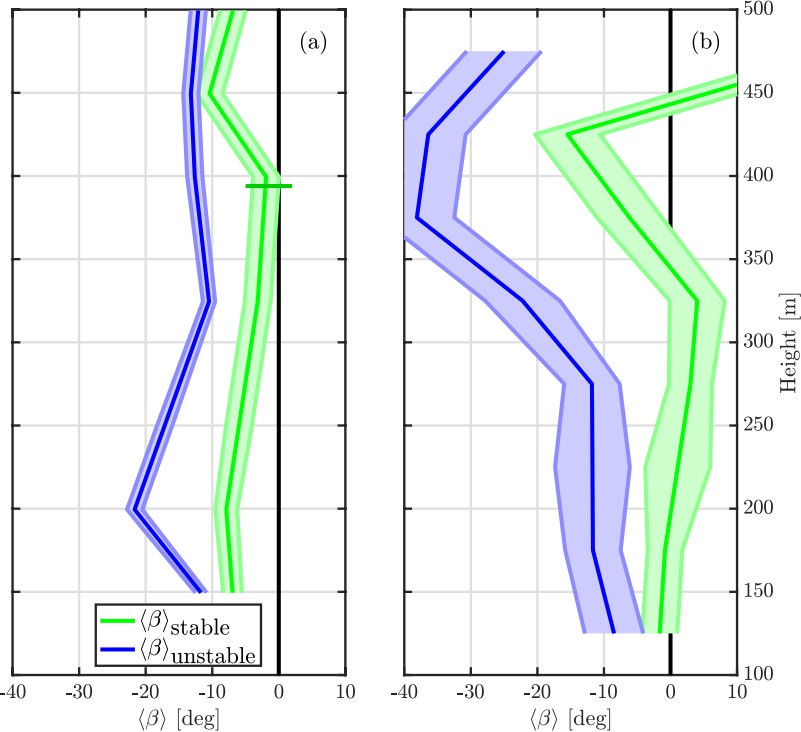

**Figure 7.** Vertical profiles of $\langle\beta\rangle$ for FINO3 (a) and Hamburg (b) under unstable (blue) and stable (green) conditions. The horizontal green line represents WRF's mean PBL height for stable conditions.

of the impact of the subgrid-scale model on the eddy fluxes and the excessive vertical wind shear within the ASL that the subgrid-scale model produces (Mirocha et al., 2018).

Over land (Fig. 8b), $\alpha$ increases with height, is always positive, and is slightly higher than the observed values at the Hamburg tower. $\beta$ is, as in the offshore case, nearly $0°$ within the first $\approx550$ m. Due to the high roughness of the onshore case, the resolved terms are less impacted by the subgrid-scale model and so $\beta$ does not show a strong departure from zero within the ASL, although there is also excessive vertical wind shear.

## 4   Discussion and Conclusions

We present novel measurements performed with a wind lidar on the FINO3 offshore platform and close to the Hamburg weather mast that show a clear misalignment between the stress vector and the vertical gradient of the mean wind vector $\beta$ up to $500\,\mathrm{m}$. Such misalignment is assumed to be zero by the PBL schemes that use $K$-theory and that are normally utilized within current numerical weather models and for wind resource assessment, e.g., the WRF model.

The observed $\langle\beta\rangle$-values increase from $\approx -5°$ at 100 m to $\approx -20°$ at 500 m at both the offshore and onshore site. During
stably stratified conditions, where larger vertical wind gradients are present, $\beta$ is still negative but larger than $-10°$ and





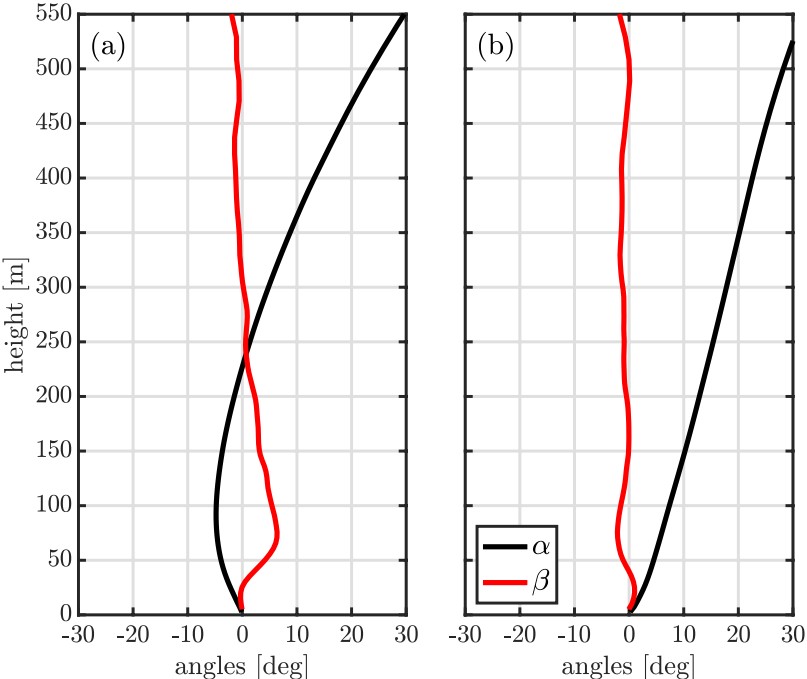

**Figure 8.** Vertical profiles of turning angles computed from the idealized neutral WRF-LES over (a) water and (b) land.

approaches zero at the PBL top. The $\beta$ has a larger magnitude for unstable conditions at all heights for both sites, also increasing with height (in magnitude) onshore. Hence, the basic assumption of $K$-theory, which we utilize to derive the eddy fluxes from the NEWA-WRF outputs holds better under stable atmospheric conditions compared to unstable conditions, as expected, due to the larger coherent structures in the latter.

Our results show that the vertical profiles of the momentum flux, which are derived from radial velocity measurements of a pulsed profiling lidar, are in agreement with those derived from the NEWA-WRF outputs. We assess the filtering effect inherent to the lidar's probe volume averaging using the Mann model and an estimation of the length scale for both sites. The turning of the stress vector relative to the mean wind vector (i.e., $\alpha$) derived from the NEWA-WRF mesoscale outputs agrees well with the lidar-derived values under both offshore and onshore conditions. This agreement is likely caused by a balance between the

wind direction and momentum fluxes biases.

     We observe an underestimation of the wind turning by NEWA-WRF at the offshore site and an opposite turning at the onshore site. We speculate that this is mainly due to a wrong surface roughness assignment (or wrong land use description) at around the Hamburg tower in NEWA-WRF. The wind direction bias is larger (in magnitude) under stable atmospheric conditions for both sites. At the offshore site, the wind direction bias is small ($\approx 1.5°$) at 100 m and increases with height, whereas the bias

decreases (in magnitude) with height at the onshore site. Sandu et al. (2013) argued that such wind turning biases in mesoscale models are the product of artificial enhancement of the mixing by turbulence closures, which in turn modifies the Ekman pumping and has a direct impact on, e.g., the ability to simulate cloud formation mechanisms.



The idealized WRF-LESs both onshore and offshore also show $\beta$ values close to $0°$ within the bulk of the PBL, further demonstrating the need to investigate, first, the conditions at which this misalignment occurs and, second, the need to better represent the PBL with numerical models. Note that the simulations of Berg et al. (2013) and Kosović and Curry (2000) used a different LES framework than our WRF-LES. A real-time WRF-LES, like performed by Schalkwijk et al. (2015), might be useful for further investigating, e.g., whether or not forcing conditions affect the behavior of $\beta$.

*Data availability.* The NEWA data are available from https://map.neweuropeanwindatlas.eu/. The FINO3 met mast and buoy data are available from https://www.bsh.de/. The Hamburg weather mast data are available from https://wettermast.uni-hamburg.de/. The LES and lidar data are available upon request.

*Author contributions.* Conceptualization and resources: J.M., P.S. and A.P. Data curation: P.S. Methodology: P.S., A.P. and J.M. Formal analysis and visualization: P.S., A.P. and J.M. Writing - original draft: P.S. Writing - review & editing: P.S., A.P. and J.M.

*Competing interests.* The authors declare no conflict of interest.

*Acknowledgements.* We acknowledge the Test and Measurements section of DTU Wind Energy for the operation and maintenance of the lidar database. This work was partly funded by the Ministry of Foreign Affairs of Denmark and administered by the Danida Fellowship Centre through the "Multi-scale and Model-Chain Evaluation of Wind Atlases" (MEWA) project.

## Appendix A: Length scale estimation

Figure A1 shows the vertical profiles of the estimated length scale $L_{MM}$ according to Eqn. (7) from observations at both sites. At the FINO3 site, where unstable conditions are predominant, the length scales decrease with increasing wind speed, partly due to the dominant neutral stability under high wind conditions. Contrarily, within the urban sector from the Hamburg tower, the length scale increases with increasing wind speeds, similarly to previous observations over flat and homogeneous land (Peña et al., 2010).





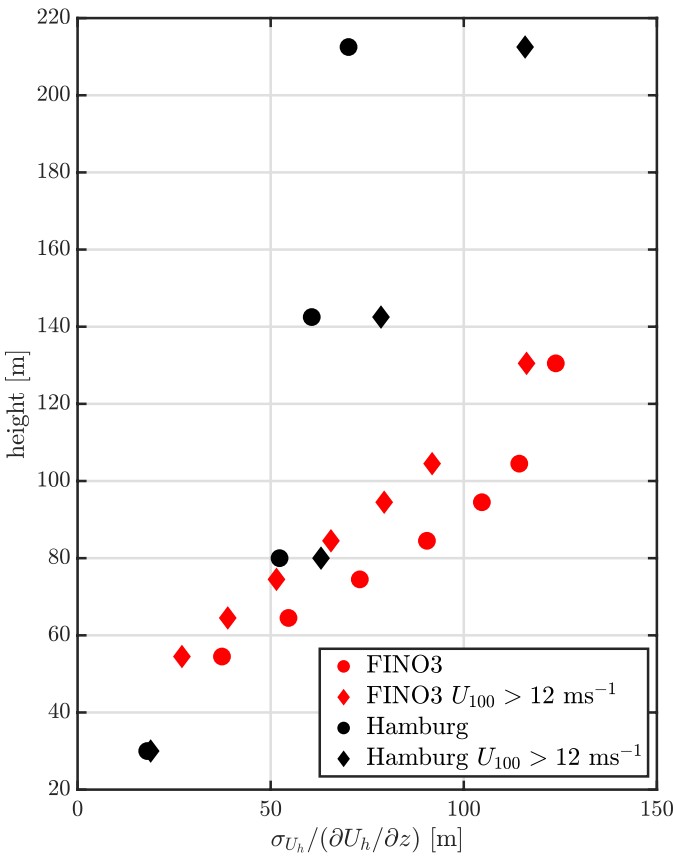

**Figure A1.** The behavior with height of the ratio of the standard deviation of the wind to the vertical wind shear under onshore (black markers) and offshore (red markers) conditions within the selected wind sector. Circles represent median profiles of all data and diamonds are median profiles for wind speeds higher than 12 m s$^{-1}$.

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
