# Peer review of "Departure from *K*-theory in the planetary boundary layer"

_Atmospheric Chemistry and Physics, 2020_

## Referee Comment (RC1) · Anonymous Referee #1 · 2 Dec 2020

This paper presents lidar measurements and NEWA-WRF simulation data for two sites, one located onshore and another offshore. The main focus is on the divergence between the turbulent fluxes and velocity vector (and its gradient) so as to link how valid K-theory, i.e. eddy diffusivity approach, is in WRF-like numerical models. Results suggest that WRF struggles to capture the wind velocity vector and turbulence statistics compared to the lidar data, which authors suggest is due to the adoption of the turbulent-diffusivity approach so as to compute the Reynolds stresses. Overall, the paper is well-structured and well-written with interesting discussions from the authors.

The reviewer finds two major issues with this papers, being the first and foremost important the self-plagiarism with the paper Santos et al. 2020 JPCS: 1618. This this cite work authors present already the FINO3 offshore measurements and WRF results

(which is somehow OK) but the text is almost a copy and paste from this other paper. And this leads to the next issue which is the limited contribution to the field in the format of a journal paper and/or lack of sufficient validation to confirm that K-theory is not good enough. Most of the WRF LES results, e.g. this inability of WRF to capture the wind dynamics in the PSL, have been already reported by other researchers as authors cite those works. At the end of Section 3, authors suggest that sub-grid-scale effects are playing a role on the LES results (again citing papers that have shown this already), but this can be analysed refining the grid near the wall so the sgs model contributions diminish, or computing the ratio of turbulent-to-natural viscosity.

In the eyes of the reviewer, the scope of the paper is well thought but the evidences here are not enough. Authors could expand more on more sites to generalise this instead of using two limited locations. Also inferring the physical constrains of the numerical model could be explored much more to validate the main question of the unsuitability of K-theory for the PSL.

Unfortunately, the reviewer recommends this paper to be rejected.

---

## Referee Comment (RC2) · Anonymous Referee #2 · 3 Dec 2020

The manuscript examines the turning of the wind with height in the atmospheric boundary layer over land and sea, in relation to the angle $\alpha$ between the stress vector and the mean wind direction, and the angle $\beta$ between the stress vector and the vertical gradient of the mean velocity vector. Processed data from Doppler Lidar measurements are presented at two sites, one located over land near Hamburg, Germany, and one located over sea, offshore in the North Sea at the FINO3 research platform. The data show that $\beta$ is approximately between $-20°$ and $-10°$ in the range $100$–$500$ m above the surface for those two sites. When averaged over this range, $\beta$ is somewhat larger at the site over sea ($-18°$) than at that over land ($-12°$).

This range of values for $\beta$ over land near Hamburg, reported in the present manuscript, is consistent with that reported over land at the Høvsøre measurement site in Denmark

by Berg et al. (2013), though larger values were reported at the latter site. The very same data for the offshore site had already been presented by Santos et al. (2020) and large parts of the present manuscript are paraphrasing/plagiarising the work by Santos et al. (2020). The present manuscript does not offer any additional insight in relation to the observed values of $\beta$ (and $\alpha$). How do we explain that they vary from site to site the way they do?

The manuscript correctly points out that the often-used flux–gradient relationship, whereby the stress vector is proportional to the vertical gradient of the mean velocity vector through a turbulent eddy viscosity, leads to $\beta = 0$. Yet, the manuscript presents a comparison of the observed values of $\beta$ to those simulated with models using such a formulation. Not surprisingly, the simulated values of $\beta$ are close to zero and the agreement with the counterpart observed values is very poor. The present manuscript does not offer any additional insight in relation to the simulated values of $\beta$ (and $\alpha$). How do we explain that they vary from site to site the way they do?

What is to be learnt from the manuscript? The manuscript needs substantial, significant work so as to be make a contribution to scientific progress. I suggest not to further proceed with its consideration for publication in a journal in the current form.

---

## Author Comment (AC1) · 2 Feb 2021

We have received two reviews from anonymous referees, which are addressed together since their comments pertain to the same topics.

Our objective with this journal paper is to expand on the conference paper Santos et al. 2020 JPCS: 1618. Therefore, we disagree with comments related to plagiarism because many journals accept such an expansion of scope, as long as the journal paper is significantly expanded. We consider that this paper has sufficient new content, with new observations and numerical simulations that falsify the hypothesis that the vector of vertical flux of horizontal momentum and the vector of mean vertical gradient of horizontal velocity are aligned. It is worth mentioning that observed values of

such misalignment over the sea have not been reported before the original conference paper.

We do not consider a problem that the paper does not offer an explanation for the behavior of such misalignment, which evidences the departure of what is assumed by the K-theory. Should scientific results only be published when explained? We present careful observations of a phenomenon and we do not offer an explanation of it, but we still think they are of general interest and that others might explain them later.

On behalf of the authors,

Pedro Santos